# The Impact of the COVID-19 Pandemic in Postoperative Neurosurgical Infections at a Reference Center in México

**DOI:** 10.3390/antibiotics12061055

**Published:** 2023-06-15

**Authors:** José Luis Soto Hernández, Luis Esteban Ramírez González, Guadalupe Reyes Ramírez, Carolina Hernández Hernández, Natalia Rangel Torreblanca, Verónica Ángeles Morales, Karen Flores Moreno, Miguel Ramos Peek, Sergio Moreno Jiménez

**Affiliations:** 1Infectious Diseases Department, Instituto Nacional de Neurología y Neurocirugía Manuel Velasco Suárez, Tlalpan, Ciudad de México 14269, Mexico; luis.ramirez@innn.edu.mx; 2Infection Control Committee, Instituto Nacional de Neurología y Neurocirugía Manuel Velasco Suárez, Tlalpan, Ciudad de México 14269, Mexico; reyesragpe@innn.edu.mx (G.R.R.); qaro.hernandez@innn.edu.mx (C.H.H.); rangeltorreblancanatalia.leo@gmail.com (N.R.T.); 3Microbiology Laboratory, Instituto Nacional de Neurología y Neurocirugía Manuel Velasco Suárez, Tlalpan, Ciudad de México 14269, Mexico; vero_angeles_morales@yahoo.com.mx (V.Á.M.); karen.flores@innn.edu.mx (K.F.M.); 4Neurosurgery Division, Instituto Nacional de Neurología y Neurocirugía Manuel Velasco Suárez, Tlalpan, Ciudad de México 14269, Mexico; miguel.ramos@innn.edu.mx (M.R.P.); smoreno@innn.edu.mx (S.M.J.)

**Keywords:** COVID-19 pandemic, neurosurgery, post-operative complications, surgical site infections, wound infection, CSF diversion procedures, ventriculitis

## Abstract

The COVID-19 pandemic has had a major global impact on the treatment of hospitalized surgical patients. Our study retrospectively evaluates the impact of the COVID-19 pandemic at a neurosurgical reference center in Mexico City. We compared the number of neurosurgeries, the rate and type of postoperative infections, the causative microorganisms and in-hospital mortality rates in a 4-year period, from the pre-pandemic year 2019 until 2022. A total of 4150 neurosurgical procedures were registered. In 2020 the total number of surgeries was reduced by 36% compared to 2019 OR = 0.689 (95% CI 0.566–0.834) *p ≤* 0.001, transnasal/trans sphenoidal pituitary resections decreased by 53%, and spinal surgeries by 52%. The rate of neurosurgical infections increased from 3.5% in 2019 to 5.6% in 2020 (*p* = 0.002). Regarding the microorganisms that caused infections, gram positive cocci accounted for 43.5% of isolates, *Klebsiella* spp. and *Pseudomonas* spp. caused one third of the infections. No significant differences were found for in-hospital mortality nor patterns of resistance to antibiotics. The number of surgeries increased in the last two years, although the infection rate has returned to pre-pandemic levels. We observed a lower impact from subsequent waves of COVID-19 and despite an increase in the number of surgeries, the surgeries have not amounted to the full pre-pandemic levels.

## 1. Introduction

Since the first weeks of the COVID-19 pandemic, the care for hospitalized medical and surgical patients has been significantly affected, and the pandemic has continued to test the limits of health care system capacities. From the onset there was a drastic decrease in neurosurgical activities, numerous surgeries were cancelled, there was a sharp decrease in face-to-face consultations, and resident activity transitioned to videoconferencing [1]. As the pandemic progressed, social distancing measures and the use of full personal protective equipment were established, and hospital resources for neurosurgical areas were limited [2]. The implementation of these measures considerably prolonged the waiting period for scheduled patients, leading to an inevitable progression or aggravation of the most severely affected neurosurgical patients [3]. Each country faced particular challenges in the face of the COVID-19 pandemic. Reports from developed countries suggested that the pandemic conditioned a slight reduction in operative capacities, and affected mainly elective procedures [4,5]. Some centers reported an increase in the number of urgent procedures with stable complication rates, and no higher mortality rate than had been reported in the pre-pandemic period [6,7]. Developing countries with higher healthcare system stress reported a considerable impact by decreasing the number of scheduled surgeries, increasing the proportion of urgent surgeries, and increasing the proportion of emergency surgeries [8,9], while in other countries there was an increase in hospital mortality and in the cost of procedures [10]. At our institution in June 2020, a multidisciplinary team of various specialists began a review of the available literature focusing on neurosurgery and COVID-19 to establish a management consensus. A document was published in December 2020 [11] with recommendations for the evaluation of neurological and neurosurgical patients that introduced the categorical CT assessment scheme for the suspicion of pulmonary involvement of coronavirus disease, known as the CO-RADS scale [12]. The scale was used to evaluate admission chest tomographies of new patients who attended our hospital emergency department. The CO-RADS scale was available in August 2020, and later, a faster turnaround of PCR SARS-CoV-2 tests results, and the availability of rapid antigens tests for SARS-CoV-2 facilitated the identification of patients with infection and staged disease severity. These improvements also allowed hospitals to implement isolation and protection measures, as well as the referral of patients between different institutions. Our facility was not designated by the national health authorities as a COVID-19 hospital. In this study, we set out to retrospectively assess the impact of the COVID-19 pandemic on the number of surgeries performed, as well as on the incidence of neurosurgery-associated postoperative infections, infecting microorganisms, antimicrobial resistance and mortality rates. We included the period from 1 January 2019 to 31 December 2019 as the pre-pandemic period and from 1 January 2020 to 31 December 2022 as the pandemic years. This single-center, retrospective study was conducted at the Department of Neurosurgery at the Instituto Nacional de Neurología y Neurocirugía Manuel Velasco Suárez in México City, México. We have had a program for the prevention and control of health care–associated infections since 1990, with two specialized nurses dedicated to the surveillance and monitoring of surgical wounds and of postoperative patients with fever. The definitions used for infections of superficial and deep surgical wounds, organs and spaces, meningitis, ventriculitis, and infections associated with cerebrospinal fluid (CSF) shunts are based on the criteria proposed by the CDC/NHSN surveillance definition of health care–associated infections and criteria for specific types of infections in the acute care setting, published in 1988 by Horan et al. [13]. Non-neurosurgical procedures were excluded from our analysis. Prior to surgery, all patients or responsible next of kin signed an informed consent form after being informed about all the possible risks and benefits of surgery.

## 2. Results

During the study period, data from 4150 neurosurgical procedures were recorded and a total of 160 postoperative infections were registered: an overall infection rate of 3.85%. Throughout the year 2020, a national lockdown was enforced for hospitals, and outpatient clinical visits were canceled from 23 June 2020 until 22 February 2021. Figure 1 shows the total number and type of neurosurgical procedures performed each year.

The most significant reduction in the total number of surgeries was observed between the pre-pandemic year 2019 and 2020, with a relative reduction of 36% in the total number of neurosurgical procedures. (odds ratio (OR) = 0.689, 95% confidence interval (CI) (0.566–0.834, *p* ≤ 0.001). Most notable was the decrease in transnasal/trans sphenoidal and spinal surgeries, which fell to approximately one half, with a relative decrease of 53%, and 52% respectively. Likewise, in the subgroup of cerebrospinal (CSF) diversion surgeries, the reduction was 31%. 

Table 1 shows the annual number of neurosurgeries, percentage of infections, and in-hospital deaths of patients with neurosurgical infections. Variations in the number of in-hospital deaths that occurred in patients with neurosurgical postoperative infections did not show statistically significant differences. (OR = 0.9574, 95% CI 0.4846–1.897, *p* = 0.89).

Table 2 shows the subgroups of CSF diversion procedures. In the first year of the pandemic, the number of CSF diversion surgeries decreased by 71 procedures (31.2%), compared to the figure for the previous year, without statistical significance. Considering the reduction in the total number of neurosurgical procedures in the first pandemic year, CSF diversion surgeries accounted for about 20% of all surgeries performed, which can be attributed to the proportion of these procedures that are made in emergency situations. All of these procedures increased in 2021 and 2022, but along with cranioplasties, these procedures have not yet reached the numbers recorded in the pre-pandemic period. 

The types of infection identified in the study period are summarized in Table 3. Incisional and deep surgical site infections accounted for more than 40% of postsurgical infections, followed by cases of postoperative meningitis and infections associated with CSF diversion procedures. In the year 2020, an increase in surgical site infections was observed in comparison with the previous year. The rate increased from 3.5% in 2019 to 5.6% in 2020 (*p* = 0.002). Thereafter, the number of these infections decreased, and the variations in the remaining types of infections were not significant. The 15 episodes of ventriculitis that occurred during this period, even though they represent less than 10% of the total, are of particular interest due to the high mortality rate, especially given that they were caused by microorganisms with reduced susceptibility to antimicrobial agents.

Table 4 shows the microorganisms recovered in surgical wounds or in Cerebrospinal fluid (CSF). The number of isolates is greater than the total number of infections, because there were eight cases in which cultures were positive for two different microorganisms. Gram-positive agents accounted for 43% of the total, gram-negative bacteria as a group were responsible for 53% of infections, and Candida albicans was isolated in 3% of cases. Klebsiella pneumoniae and Pseudomonas aeruginosa stand out for their numerical relevance because they are pathogens associated with multiresistance and represented one third of the causative agents of post neurosurgical infections. The low number of isolates per species, as well as the number of multiresistant (MDR) and extended resistance (XDR) microorganisms did not show statistically significant differences during the study period.

From 23 March 2020 until 15 July 2020 all scheduled outpatient appointments (about 10,000) were canceled. Elective neurosurgery was canceled between April 2020 until February 2021, nevertheless we observed an increase of 30% in surgical evaluations in our emergency department [11], due to an increase in referrals from other neurosurgical services that were converted to COVID-19 hospitals. The occupancy of beds in the neurosurgery service was reduced to about 70%. In order to maintain the academic activities of the neurosurgery residents, clinical case sessions were scheduled nightly, while neuroanatomy, clinical webinars, and professor videoconferences of clinical and research topics were offered. Two senior neurosurgeons implemented telehealth outpatient consults, and between May and December 2020, they imparted 192 teleconsultations. From January 2021 onwards, the number telehealth outpatient consults reduced to 15, when face-to-face visits restarted, and those were all telehealth visits given in that year as elective neurosurgery resumed.

## 3. Discussion

### 3.1. Reduction of Neurosurgical Procedures

In our country during the initial phase of the pandemic, the most significant problem was the scarcity of diagnostic tests, which were only carried out in a national reference center and thus caused a delay of several days in reporting the results. Gradually, public and private laboratories were certified and facilitated the testing in other institutions. In August 2020, the publication the CO-RADS scale [12] was an important tool for evaluating patients in the emergency room. However, the cancellation of admissions for elective neurosurgery and the outpatient clinics activities ordered by national health authorities reduced the number of surgical procedures. By adding the information from the chest CT scans to the level of neurosurgical urgency it was possible to define whether to proceed to surgery immediately, if the case required it, or if possible, to wait for the PCR results tests in respiratory secretions. During the year 2021, the national vaccination campaign against SARS-CoV-2, together with a faster turnaround time of results of PCR tests, availability and consistent adherence to the use of PPE, made possible the return of medical and paramedical personnel to the hospital, restoring the productivity of the neurosurgery service to a large extent. However, at the end of the year 2022, full productivity still had not resumed for spinal surgery, cranioplasty or transnasal/trans sphenoidal surgery. In the early pandemic period, a report from an academic neurosurgery department [1], indicated that overall, the neurosurgical operative volumes decreased from 360 total cases in April 2019 to 112 projected total cases for April 2020, representing a reduction of 68.89%. This occurred in a center in which a large number of scheduled surgeries were canceled, which seriously affected neurosurgery education, which largely shifted toward video-conferencing sessions rather than in-person sessions [1,2,3]. Jankovic et al. [7], compared a 15-month pre-pandemic period with the first 15 months of the COVID-19 pandemic, and analyzed the total number of spinal surgeries and postoperative complications at a neurosurgical center in Germany. They found that the total number of procedures and complications were similar, but observed that the number of emergency spinal surgeries increased by 10.2% in the pandemic period, and the number of patients transferred from other hospitals under emergency conditions increased by 14%. They did not find changes in the frequency of post-surgical infections (2.4% pre-pandemic vs. 1.9%, in the pandemic periods), or in the species of causative microorganisms. Hussein et al. [5], analyzed the total number of neurosurgical procedures performed in a reference center in Ireland in 2019, 2020 and 2021. They saw a greater impact in the first month of the pandemic and in the first quarter of 2021. They also found a greater reduction of transnasal/trans sphenoidal pituitary surgeries during the pandemic, at 36% and 44% in the years 2020 and 2021, respectively, which they attributed to the potential generation of aerosols during these surgical approaches, forcing the use of complete PPE in the surgical rooms. They also highlighted a peak resource utilization in the first quarter of 2021 in preparation for a new wave of COVID-19 cases, decreasing ICU and general beds at their institution. By contrast, a report that aimed to determine how the first wave of the COVID-19 pandemic affected outcomes for all neurosurgical patients in a single hospital in Birmingham UK [14], in which the outcomes of patients undergoing neurosurgery during the height of the pandemic were compared against a matched cohort from prior to the COVID-19 outbreak with primary outcomes of 30-day mortality and postoperative pulmonary complications, found no statistically significant difference in mortality nor pulmonary complications, and the median length of stay was 4.5 days (interquartile range (IQR) 2.0–10.3) for the pandemic cohort and 6.0 days (IQR 2.0–18.0) for the matched cohort. The authors commented that the grade of primary surgeon was significantly more senior, which was likely to have negatively impacted training opportunities for junior surgeons. Also, in the first months of the COVID-19 pandemic, when Soriano Sánchez et al. [15], used an internet-based survey among presidents and members of the societies of the Latin American Federation of Neurosurgical Societies (FLANC), information was provided by 21 countries. They obtained 486 completed questioners in a single day. Responses from society presidents informed the researchers of their having suspended regular activities and deferring local scheduled congresses, fourteen reported mandatory isolation by their government and four instituted a telemedicine project. Four-hundred and eighty-six neurosurgeons reported an average reduction of 79% in their neurosurgical praxis. In a systematic review and meta-analysis aiming to quantify the reduction in neurosurgical operative volume and describe the impact of these trends on neurosurgical residency training, Kuo et al. [16], included 49 studies that met their inclusion criteria, published between December 2019 and October 2022, 12 of which were survey-based studies. The case volume of elective surgeries and non-elective procedures decreased by 70.4% and 68.2%, respectively, between pre-pandemic and pandemic periods. Single-center studies with a similar design from several countries reported percentage reductions in neurosurgeries: Munda et al. [17] from Ljubljana Slovenia observed an 11.2% reduction. Permana et al. [18] from Indonesia, recorded a 68% reduction in spine surgery. ElGhamry et al. [19] observed a 21% reduction in neurosurgical procedures in Newcastle, UK. Chau et al. [20] from Kentucky, USA, indicated that their neurosurgical cases dropped 55% from their baseline. Idowu et al. [9] in Lagos, Nigeria, found a 41.8% reduction in surgical procedures during the pandemic period. The reduction in caseloads had caused concerns among residents and program directors in regard to the diminished clinical exposure, financial constraints, and their mental well-being. Some positives highlighted were rapid adaptation to virtual educational platforms and increasing time for self-learning and research activities.

### 3.2. Changes in Mortality Rates after Neurosurgery during the Pandemic Period

In our study, the hospital mortality rates of patients undergoing neurosurgery did not show significant differences between the pre-pandemic year and the three years of the pandemic period. In some countries that have made reports in this regard, differences are found. Sander et al. [4], in a retrospective single center study concerning neurosurgical care at a university hospital in Germany, compared a pre-pandemic period of five and a half months with a similar period in the second wave of the COVID-19 pandemic. They found a reduction of around 11% in the number of patients and 12% in the number of surgeries in the pandemic period. Regarding mortality, they reported 4.32% in the pre-pandemic period against 5.05% in the pandemic period patient group. No significant differences in complication rates and unplanned readmissions were observed. They attributed their findings to the availability of resources and the high-quality processes that existed even before the pandemic. An increase in mortality rates was observed in several studies: Karimov et al. [21] made a comparison between a period of six pre-pandemic months to six pandemic months in a hospital in Turkey, and identified a relative reduction of 18% in the total number of neurosurgical procedures and an increase in overall mortality rates from 6.8% to 9.6% (*p* = 0.03) between the pre-pandemic and pandemic periods. The neurosurgery residents and academics of their center were unsatisfied with the decrease in the number of operations, training, and research productivity, and believe that the pandemic and restrictions negatively affected the health system and people’s access to healthcare. They pointed out the importance of not arriving late to the hospital despite restrictions, which has a deleterious effect on the survival of critically ill patients. De Macêdo Filho et al. [10] analyzed 11 neuro-oncology surgical procedures from a neurosurgery information health system in Brazil, between February and July of 2019 and during same period in 2020. There was a 21.5% decrease in the number of procedures. The mortality rate during hospitalization in neurosurgical cases increased by 22.26% during the pandemic period. They commented that the reduction in the number of elective neurosurgical procedures to minimize the exposure of fragile patients to COVID-19 and the prioritization of patients requiring early surgical intervention lowered the number of neurosurgeons, operating rooms, and neurosurgery beds. They concluded that in Brazil, the COVID-19 pandemic increased neurosurgical mortality, decreased neurosurgical hospitalizations and considered their study a stark example of the effects of the COVID-19 pandemic on neurosurgical care in settings with limited resources and access to care. Poon et al [22], in an international observational multicenter cohort study, assessed treatment pathways and perioperative events in patients undergoing surgery for a tumor during the COVID-19 pandemic. They recruited adults who underwent surgery for an intracranial tumor during the period of January to August 2020. Patients were categorized by location and income group as high [HIC], upper-middle [UMIC], and low- and lower-middle [LLMIC]. One of the main outcomes was 30-day mortality. Using multilevel logistic regression, they estimated the association between income groups and mortality. Within 30 days after surgery, 39 (3.8% of) patients died. In the multivariable model, LLMIC was associated with increased mortality (odds ratio 2.83, 95% credible interval 1.37–5.74) compared to HIC, leading the authors to conclude that the disparity in mortality rates between countries with varying incomes warrants further examination to identify any modifiable factors.

### 3.3. Impact on Training of Neurosurgical Residents Worldwide, Telehealth Visits and Satisfacton

The studies that evaluated how COVID-19 affected the admission of neurosurgery residents among academic neurosurgery departments were regional database reviews or surveys. Dokponou et al. [23] compared neurosurgery resident training and admission in low middle-income countries (LMICs) and high-income countries (HICs), from 2019 to 2021, and found 58 studies; 42 (72.4%) were conducted in HIC and 16 (27.6%) in LMIC. New resident admission was canceled in thirteen HIC and in four LMIC from 2019 to 2021. Learning modalities changed to video conferencing, and neurosurgery was restricted to emergency cases in 79.6% of studies, with 12.2% of these being elective cases. Tomlinson et al. [24] wrote an editorial early in the pandemic, questioning if the COVID-19 pandemic made it necessary to re-examine neurosurgical training models when social distancing was the most effective measure for limiting transmission and medical students had been either banished from the wards or ushered into the workforce. Although there is no substitute for time in the operating room, residency programs migrated components of the training curriculum to online, and the editorial suggested that lessons learned from the COVID-19 pandemic will invigorate researchers working to bring high-fidelity surgical simulators into neurosurgical education.

In our institution we had a reduced number of telehealth visits during 2020, mostly due to difficult connectivity or lack of communication equipment among our patients. Literature presents contrasting data in this respect. A survey among members of the Congress of Neurological Surgeons [25], to which 363 members responded, indicated that on average 60% of consultations had been converted to telehealth consults. 70% of responders expected telemedicine to increase urgent appointments but 56% thought that it could decrease the quality of the relationship between practitioners and patients. The limitations of telemedicine described included the impossibility to perform physical examinations (42%), the inability of patients to use technology (24%) and to work with elderly patients (20%). Hopkins et al. [26] reviewed new outpatient in-person visits (IPVs) and telehealth visits (THVs) and found that THVs did not increase the duration of follow ups. Porche et al. [27] evaluated patient satisfaction scores of telemedicine clinic visits in neurosurgery in comparison to in-person visits in neurosurgical clinics in Florida, USA. The response rates were 20% (97 responders) for telemedicine visits and 19% (589 responders) for in-person visits. Patient overall satisfaction scores were higher with telemedicine visits compared to in-person, corrected for care provider differences. Dehora et al. [28] found in an online survey with 176 responses from the Indian subcontinent that of all respondents, 46% were practicing restricted outpatient services, more so in public institutions, which also had a higher incidence of tele-outpatient services: 26% vs. 17% in private practice. Menlibayeva et al. [29] reviewed the neurosurgical practice in Kazakhstan during a country-wide lockdown in July 2020 and found a decrease in consultations (65.34%) and surgeries (56.55%) by all neurosurgeons, regardless of city type. However, neurosurgeons attended online educational courses during the pandemic, mainly in major cities. Ashry et al. [30] evaluated the impact of COVID-19 on neurosurgical residency programs in five tertiary medical centers in Egypt via a survey sent to neurosurgery residents. They detected a reduction in surgical cases, inpatient services, and working hours per week during the pandemic compared to the pre-pandemic era.

### 3.4. Limitations and Strengths of Our Study 

Our study presents data from a single center via a retrospective design, which prevents our data from being applied to other centers. One strength of our study is the inclusion of microbiological findings, which are not found in most similar studies on postoperative complications in neurosurgery during the COVID-19 pandemic. For us, it is important to point out that the interaction between infectious disease services, infection prevention and control programs, and the analysis of microbiological data, allow for targeted use of antibiotics and possibly improved results in neurosurgical centers, even in demanding circumstances such as the COVID-19 pandemic.

## 4. Materials and Methods

Study design and population. This is a single center retrospective cohort study. 

Inclusion criteria. All adult patients that underwent a neurosurgical procedure between 1 January 2019 until 31 December 2022 at the Instituto Nacional de Neurología y Neurocirugía Manuel Velasco Suárez in Mexico City were included. All patients were 18 years of age or older, and all recorded information was anonymized. Data were obtained from records in the neurosurgery service database. Patients referred from other institutions with a postoperative neurosurgical infection were not included as new infectious cases, but the procedures required for their treatment were included.

Exclusion criteria. Surgical interventions of ophthalmology, otoneurology, and endovascular therapy services were excluded. The neurosurgical procedures were grouped according to the final diagnosis issued by the neurosurgeon in the surgical note.

Postoperative infections. The detection of postoperative infections was carried out during the intrahospital surveillance of surgical wounds and episodes of fever by the nurses of the committee for prevention and control of health care associated infections, and cultures of secretions from wounds or cerebrospinal fluid, if indicated, were obtained via lumbar puncture or ventricular drainage in patients with suspected postoperative meningitis. Positive cultures were interpreted along with the CSF cytochemical study to define the presence of infection. 

Classification of infections. As previously indicated, each episode of infection was classified according to the recommendations of Horan et al. [13]. Infectious diseases physicians attend the outpatient clinic weekly to monitor discharged patients following postoperative infections. 

Infecting microorganisms. Information on microorganisms recovered in infections and their susceptibility to antibiotics was recorded from the institutional microbiology laboratory system, with special interest in multi-resistant microorganisms. Multidrug-resistant (MDR) was defined as acquired non-susceptibility to at least one agent in three or more antimicrobial categories, and extensively drug-resistant (XDR) was defined as non-susceptibility to at least one agent in all but two or fewer antimicrobial categories (i.e., bacterial isolates remain susceptible to only one or two categories) [31]. 

Hospital deaths. In-hospital deaths of patients with postoperative infection were recorded after detailed review of the medical records of each patient identified with a postoperative infection. To reduce collection bias, each case with postoperative infection was reviewed independently by two physicians and a supervising nurse and discussed until consensus was reached. 

The Manuel Velasco Suarez National Institute of Neurology and Neurosurgery in Mexico City is a reference and teaching center for neurological and neurosurgical disorders. Around 1300 neurosurgical procedures are performed per annum. Most craniotomies are elective for the resection of gliomas, meningiomas and pituitary adenomas. About one half of vascular surgeries for subarachnoid hemorrhages and cerebrospinal fluid (CSF) diversion surgeries are performed as emergency procedures. Cranial trauma and polytrauma patients are not regularly admitted at our institution, except for occasional subdural hematomas (other centers in our city admit trauma patients). Two specialized nurses for the prevention and control of health care–associated infections perform surveillance activities and visit the forty-two beds in the neurosurgery ward, nine bed recovery room, and nine bed intensive care unit daily, and keep records of patients with surgery and invasive procedures. They suggest that nurses and residents obtain samples for cultures, and subsequently inform of culture results. A written monthly report of infections is made to the institutional authorities, and the Prevention and Control Committee for Health Care-Associated Infections meets quarterly to present and discuss infection trends. Antibiotic surgical prophylaxis is performed in all cases of craniotomies, spinal cord surgeries, and transnasal/trans sphenoidal approaches for pituitary adenomas, and in CSF diversion surgeries with three doses of ceftriaxone every 12 h, beginning during anesthetic induction. The decision to prescribe additional antibiotics for patients that remain intubated for more than 24 h postoperatively, or are in critical condition, is discussed by the treating physicians and the infectious disease service. 

Statistical Analysis. Statistical analysis was performed with IBM SPSS Statistics 28.0 software (IBM, Armonk, New York, NY, USA), and CDC Epi Info™ public domain soft-ware package version 7.2. Data were described in terms of frequencies (number of cases) and percentages and compared using two-sided *p* values. The associations between continuous variables were examined using the *t*-test for normal distribution and the Mann–Whitney U test or Kruskal Wallis test for variables without normal distribution. Categorical variables were compared by employing the Chi-square (χ^2^) test. Continuous variables were described using mean and median values, while categorical variables were described with counts and frequencies. A two-tailed *p* value < 0.05 was considered to be statistically significant.

## 5. Conclusions

Our study showed a significant impact in the first year of the COVID-19 pandemic in our center, with a reduction in the total number of surgeries and an increase in the rate of postoperative infections. The availability of rapid antigen tests for SARS-CoV-2, along with the vaccination of health personnel, allowed for a rapid recovery in the number of surgeries in 2021. However, in transnasal/trans sphenoidal approaches for pituitary adenomas and spinal surgeries, the number of pre-pandemic procedures has not fully recovered. The review of similar studies showed different scenarios related to the socioeconomic level of different countries, and that most neurosurgery services in hospitals were converted into COVID-19 centers. This created a situation in which the redistribution of the flow of patients in periods of aggravation of the pandemic reduced the availability of general beds, intensive care, and recovery rooms for neurosurgical patients in critical condition. The generalized and forced lockdowns, with police and military participation in some countries, impacted patients’ timely access to medical care for serious diseases, increased the risk of complications and increased mortality rates. There are not enough studies that include information about bacteriological data and antibiotic susceptibilities to COVID-19, so it is not possible to find definite trends, nor elements of comparison. The reserve of hospital supplies to prevent long periods of scarcity, a robust response plan for unexpected events, and timely access to health services can all reduce the impact of unpredictable events such as this pandemic.

## Figures and Tables

**Figure 1 antibiotics-12-01055-f001:**
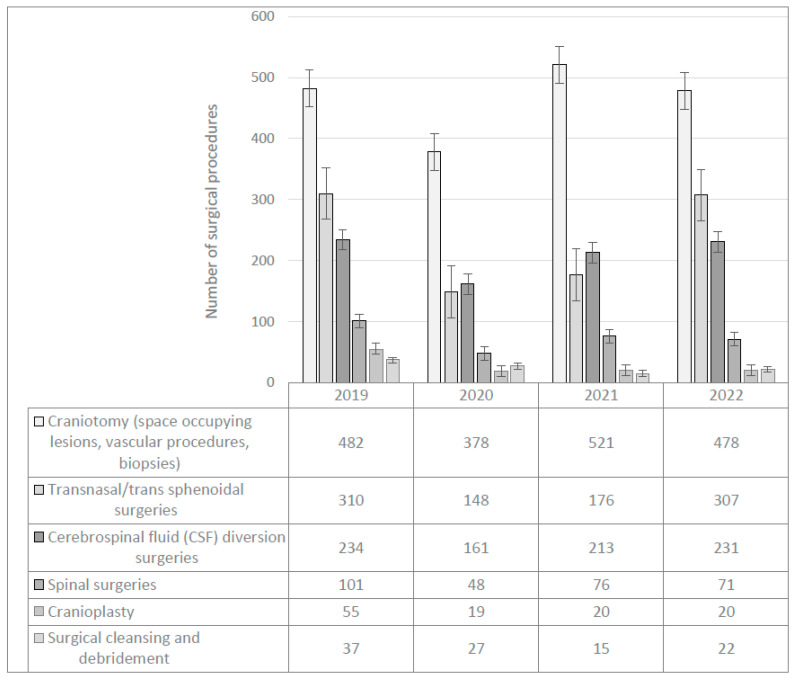
Number of surgical procedures per year.

**Table 1 antibiotics-12-01055-t001:** Number of neurosurgeries, postoperative infections and hospital deaths by each study tear.

Year	2019	2020	2021	2022
Number Neurosurgeries/year	1219	781	1021	1129
Infections number (%)	39 (3.19)	44 (5.63)	42 (4.11)	35 (3.10)
Hospital deaths (%)	7 (0.57)	6 (0.76)	11 (1.07)	11 (0.97)

**Table 2 antibiotics-12-01055-t002:** Subgroup analysis of CSF diversion surgeries.

Year	2019	2020	2021	2022
Ventriculoperitoneal shunt (VPS) initial placement	124	83	112	142
Ventriculostomy for acute hydrocephalus	35	23	43	39
Ventriculostomy for intracranial bleeding	25	21	22	19
Change from ventriculostomy to VPS	19	12	11	3
Change from VPS to ventriculostomy	9	4	16	16
Ventriculostomy after VPS removal due to infection or exchange	22	18	9	12
Total	234	161	213	231

**Table 3 antibiotics-12-01055-t003:** Type of infection by year, and total and percentage of each type of infection.

Year	2019	2020	2021	2022		
Type of Infection					Total	(%)
Surgical site infections *	15	25	17	12	69	43.1
Postoperative meningitis	10	10	6	12	38	23.8
CSF shunt infection (SSI-meningitis)	7	7	11	8	33	20.6
Ventriculitis	5	1	6	3	15	9.4
Postoperative bone flap osteomyelitis	1		2		3	1.9
Postoperative abscess	1	1			2	1.3
Total	39	44	42	35	160	100

* superficial incisional, and deep incisional.

**Table 4 antibiotics-12-01055-t004:** Microorganisms recovered in cultures of surgical wounds and CSF.

Year	2019	2020	2021	2022	Total	(%)
Microorganisms						
Coagulase negative staphylo-coccus	11	11	16	8	46	27.4
*Staphylococcus aureus*	4	4	5	5	18	10.7
*Enterococcus* species	3	3	1	2	9	5.4
*Klebsiella pneumoniae*	4	10	6	10	30	17.9
*Pseudomonas aeruginosa*	9	5	2	9	25	14.9
Other gram negative bacilli	7	6	11	11	35	20.9
*Candida albicans*	2	1	1	1	5	2.9
Total					168 *	
MDR and XDR ** bacteria	9	9	4	4		

* Two microorganisms isolated in eight cultures; ** MDR (multidrug-resistant organisms) were defined as acquired non-susceptibility to at least one agent in three or more antimicrobial categories; XDR (extensively drug-resistant) was defined as non-susceptibility to at least one agent in all but two or fewer antimicrobial categories.

## Data Availability

Databases supporting the results can be provided upon request.

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
