# Peer review of "The Impact of the COVID-19 Pandemic in Postoperative Neurosurgical Infections at a Reference Center in México"

_antibiotics, 2023, doi:10.3390/antibiotics12061055_

Round 1

Reviewer 1 Report

The manuscript is well-written, and I have no significant objections regarding the Introduction, Materials and Methods and Results sections.

Regarding the Discussion section, I would suggest reducing the size and merging the sections "3.3. Impact on training of neurosurgical residents worldwide" and "3.4. Telehealth visits, telehealth satisfaction", as the results addressed in these sections are not directly related to the results from this manuscript.

I have no other objection to the manuscript, and I think that it can be accepted for publication.

Author Response

Reviewer 1. We greatly appreciate your comments on our manuscript. Regarding the discussion section, we appreciate and accept your suggestion to shorten sections 3.3 impact on training of neurosurgical residents and 3.4 telehealth visits and telehealth satisfaction, and merge them into a single paragraph by combining their titles. Certainly, these themes are not in the main objectives of our study, however both aspects were as important for our institution as they were for most neurosurgical centers in other parts of the world, and the proof is found in the extensive literature on these issues, that continues to be published. We believe that they make our work more complete.

Reviewer 2 Report

The author tried to present the impact of COVID-19 pandemic in patients having that postoperative neurosurgical infections 1 center of México. It was a very informative study and can be use as a guideline.  

I have some suggestions regarding the manuscript.

-Please add more background data in the heading introduction to strong the rationale of the study as currently there are very few studies mentioned.

-in the introduction author mentioned 5 reference in the first half of paragraph. its not a good approach to do so.

-results are good but my suggestion is to draw some graph to make the manuscript good. as graphical representation is a good and easy way to inform the readers.

-mention strengths of the study in a separate heading like limitation and also how do you reduced the biases.

-regarding the method section, its not a good approach in a research paper to mention method in one paragraph. Follow the guidlines of research paper and add sub heading like study site, study population, inclusion/exclusion criterias, etc......

Author Response

Reviewer 2 response.

Thank you very much for your comments on our manuscript.

-Please add more background data in the heading introduction to strong the rationale of the study as currently there are very few studies mentioned.

-in the introduction author mentioned 5 references in the first half of paragraph. Its not a good approach to do so.

We have rewritten the introduction, and separated the references to point out specific aspects of each one. In total the introduction now has 10 references, (lines 35 to 52), and we consider that now the rational bases of the study are clearer

-results are good but my suggestion is to draw some graph to make the manuscript good. as graphical representation is a good and easy way to inform the readers.

Since a fundamental aspect of our study is the impact of the COVID-19 pandemic on the reduction in the number of neurosurgeries in the first pandemic year, a vertical bar chart was added to Table 1, turning it into Figure 1. It clearly shows with error bars the information, the associated table accurately gives the numbers.

For the subsequent tables, we consider that the graphs that would be generated were not visually appealing,  and did not provide significant details to complement the information, so they were left unchanged.

-mention strengths of the study in a separate heading like limitation and also how do you reduce the biases. We reduce the risk of bias, by collecting data of posotoperative infections in the initial monthly nursing report, reviewed by infectious disease physicians, and in the independent review of cases per year at the time of concentrating the complete information for the study period.

The section 3.4. was re-named Limitations and strengths of our study. 

-regarding the method section, its not a good approach in a research paper to mention method in one paragraph. Follow the guidelines of research paper and add sub headings like study site, study population, inclusion/exclusion criteria, etc...

The methods section. Lines 335 to 368,  was completely rewritten in order to adhere to the suggested indications.

Round 2

Reviewer 2 Report

Authors addressed my queries.